# CarRS Two-Component System Essential for Polymyxin B Resistance of *Vibrio vulnificus* Responds to Multiple Host Environmental Signals

Duhyun Ko,[a]* Dayoung Sung,[a] Tae Young Kim,[a]§ Garam Choi,[a]◇ ⓘ Ye-Ji Bang,[b,c,d] ⓘ Sang Ho Choi[a,e,f]

[a]National Research Laboratory of Molecular Microbiology and Toxicology, Department of Agricultural Biotechnology, Seoul National University, Seoul, Republic of Korea
[b]Department of Biomedical Sciences, College of Medicine, Seoul National University, Seoul, Republic of Korea
[c]Department of Microbiology and Immunology, College of Medicine, Seoul National University, Seoul, Republic of Korea
[d]Institute of Infectious Diseases, Seoul National University College of Medicine, Seoul, Republic of Korea
[e]Department of Agricultural Biotechnology, Center for Food and Bioconvergence, Seoul National University, Seoul, Republic of Korea
[f]Department of Agricultural Biotechnology, Research Institute of Agriculture and Life Science, Seoul National University, Seoul, Republic of Korea

Duhyun Ko and Dayoung Sung contributed equally to this work. Author order was determined on the basis of seniority.

**ABSTRACT** Enteropathogenic bacteria express two-component systems (TCSs) to sense and respond to host environments, developing resistance to host innate immune systems like cationic antimicrobial peptides (CAMPs). Although an opportunistic human pathogen *Vibrio vulnificus* shows intrinsic resistance to the CAMP-like polymyxin B (PMB), its TCSs responsible for resistance have barely been investigated. Here, a mutant exhibiting a reduced growth rate in the presence of PMB was screened from a random transposon mutant library of *V. vulnificus*, and response regulator CarR of the CarRS TCS was identified as essential for its PMB resistance. Transcriptome analysis revealed that CarR strongly activates the expression of the *eptA*, *tolCV2*, and *carRS* operons. In particular, the *eptA* operon plays a major role in developing the CarR-mediated PMB resistance. Phosphorylation of CarR by the sensor kinase CarS is required for the regulation of its downstream genes, leading to the PMB resistance. Nevertheless, CarR directly binds to specific sequences in the upstream regions of the *eptA* and *carRS* operons, regardless of its phosphorylation. Notably, the CarRS TCS alters its own activation state by responding to several environmental stresses, including PMB, divalent cations, bile salts, and pH change. Furthermore, CarR modulates the resistance of *V. vulnificus* to bile salts and acidic pH among the stresses, as well as PMB. Altogether, this study suggests that the CarRS TCS, in responding to multiple host environmental signals, could provide *V. vulnificus* with the benefit of surviving within the host by enhancing its optimal fitness during infection.

**IMPORTANCE** Enteropathogenic bacteria have evolved multiple TCSs to recognize and appropriately respond to host environments. CAMP is one of the inherent host barriers that the pathogens encounter during the course of infection. In this study, the CarRS TCS of *V. vulnificus* was found to develop resistance to PMB, a CAMP-like antimicrobial peptide, by directly activating the expression of the *eptA* operon. Although CarR binds to the upstream regions of the *eptA* and *carRS* operons regardless of phosphorylation, phosphorylation of CarR is required for the regulation of the operons, resulting in the PMB resistance. Furthermore, the CarRS TCS determines the resistance of *V. vulnificus* to bile salts and acidic pH by differentially regulating its own activation state in response to these environmental stresses. Altogether, the CarRS TCS responds to multiple host-related signals, and thus could enhance the survival of *V. vulnificus* within the host, leading to successful infection.

**KEYWORDS** *Vibrio vulnificus*, two-component system, polymyxin B resistance, transcriptome analysis, host environmental signals

Address correspondence to Ye-Ji Bang, yeji.bang@snu.ac.kr, or Sang Ho Choi, choish@snu.ac.kr.

*Present address: Duhyun Ko, Institute of Infectious Diseases, Seoul National University College of Medicine, Seoul, Republic of Korea.

§Present address: Tae Young Kim, Molecular Diagnostics Research Group, Bioneer Corporation, Daejeon, Republic of Korea.

◇Present address: Garam Choi, Center for Immunity and Inflammation, Rutgers New Jersey Medical School, Newark, New Jersey, USA.

The authors declare no conflict of interest.

For successful survival within the host, pathogenic bacteria should recognize their environments and elaborately regulate the expression of genes involved in the defense against the host immune system. Accordingly, pathogenic bacteria have evolved two-component systems (TCSs) to sense and respond to various host environments (1, 2). The TCS, consisting of a sensor kinase and a response regulator, transduces signals through phosphorelay reaction. When activated by specific environmental signals, the sensor kinase autophosphorylates its conserved His residue and then transfers the phosphoryl group to a conserved Asp residue of the response regulator. Generally, the phosphorylated form of the response regulator is considered an active form able to modulate the expression of its target genes (3).

During infection, enteropathogenic bacteria inevitably encounter several barriers, such as gastric acid, bile, and cationic antimicrobial peptides (CAMPs), raised by the host innate immune system (4–6). CAMPs, including $\alpha$-defensin and cathelicidin, are produced by the host epithelial cells, and their production is highly induced upon bacterial infection (6–8). CAMPs bind to negatively charged lipopolysaccharides (LPSs) of bacteria and lead to their lysis. Bacteria modify their LPS structures by adding positively charged moieties like phosphoethanolamine and 4-amino-4-deoxy-L-arabinose to prevent the binding of CAMPs to their surface. Another strategy for bacteria to resist CAMPs is associated with efflux pumps that extrude the antimicrobial peptides in an energy-dependent manner (9).

Polymyxin B (PMB) is an antimicrobial peptide and possesses a mechanism of action similar to that of CAMPs. Although PMB is one of the last treatments for Gram-negative bacteria, some bacterial species have been reported to be intrinsically resistant to the antimicrobial agent (10, 11). Especially, *Vibrio cholerae* has been reported to show resistance to PMB, as well as CAMPs, through a CarRS (also known as VprAB) TCS (12, 13). The CarRS TCS develops resistance to the antimicrobial peptides by activating the expression of *almEFG* which encodes a unique LPS modification system of *V. cholerae* (12, 14). Meanwhile, a recent study revealed that the CarRS TCS senses $\alpha$-defensin as a host-related signal to facilitate the production of virulence factors (15). Accordingly, the CarRS TCS is an important signal transduction system contributing to the successful pathogenesis of *V. cholerae*, as well as its PMB resistance.

A Gram-negative human pathogen *Vibrio vulnificus* commonly contaminates oysters and causes a range of foodborne diseases from gastroenteritis to life-threatening septicemia (16–18). Similarly to *V. cholerae*, *V. vulnificus* also shows intrinsic resistance to PMB (19, 20). However, the mechanism by which *V. vulnificus* develops the PMB resistance has not yet been characterized in detail. In the present study, *V. vulnificus* open reading frame (ORF) encoding a homolog of *V. cholerae* CarR was identified by a transposon-tagging method in an effort to search the factors involved in the resistance to PMB. Genetic analyses revealed that *V. vulnificus* CarR contributes to the PMB resistance and activates the expression of the *eptA* and *tolCV2* operons which are involved in LPS modification and efflux of antibiotics, respectively. CarR also positively autoregulates the expression of the *carRS* operon. Among the CarR regulon, the *eptA* operon plays a major role in the CarR-mediated PMB resistance. The response regulator CarR is phosphorylated by the sensor kinase CarS, and the phosphorylation is strictly required for the regulation of the *eptA* and *carR* expression that leads to the PMB resistance. CarR directly and specifically binds to the upstream regions of the *eptA* and *carRS* operons, but its DNA-binding ability is independent of phosphorylation. Notably, the CarRS TCS alters its own activation state in response to various environmental stresses, such as PMB, divalent cations including $Mg^{2+}$ and $Ca^{2+}$, bile salts, and pH change, that *V. vulnificus* may encounter in the infected host. Furthermore, CarR significantly affects the resistance of *V. vulnificus* to bile salts and acidic pH among the stresses, as well as PMB. Altogether, this study suggests that the CarRS TCS, essential for the PMB resistance, responds to host environmental signals, leading to successful survival of *V. vulnificus* during infection.

## RESULTS

**The response regulator CarR contributes to the PMB resistance of *V. vulnificus*.**
Using a transposon-tagging method, the VVMO6_RS15995 gene was identified from a

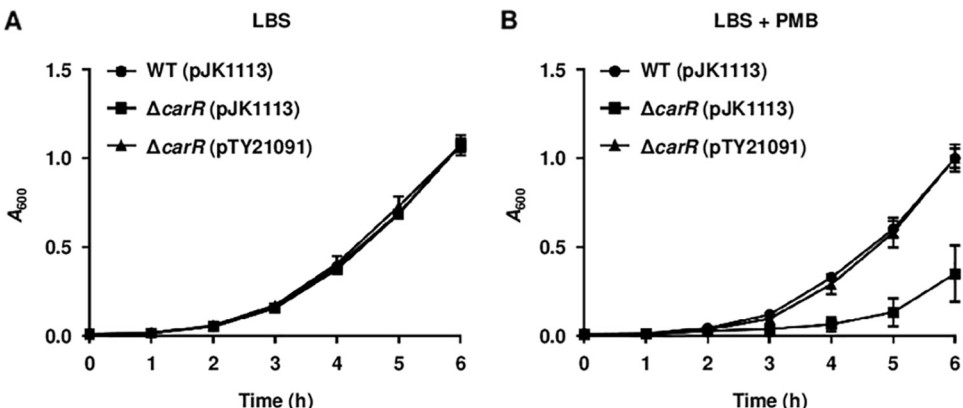

**FIG 1** Growth of *V. vulnificus* upon exposure to PMB. (A and B) The growth of the *V. vulnificus* strains was monitored at 1 h intervals in LBS containing 1 mM arabinose and 100 $\mu$g mL$^{-1}$ kanamycin supplemented without PMB (A) or with PMB (10 $\mu$g mL$^{-1}$) (B). Error bars represent the standard deviations (SD) from three independent experiments. WT (pJK1113), wild type; $\Delta carR$ (pJK1113), *carR* mutant; $\Delta carR$ (pTY21091), complemented strain.

mutant of *V. vulnificus* that exhibited a decreased growth rate in the presence of PMB than a wild type. A protein encoded by VVMO6_RS15995 is predicted to contain an N-terminal receiver domain and a C-terminal DNA-binding domain. As the protein shows sequence homology with the response regulator CarR of *V. cholerae*, VVMO6_RS15995 was designated as *carR* of *V. vulnificus*. Additionally, the VVMO6_RS15990, which is located immediately downstream from *carR*, is predicted to encode a homolog of the sensor kinase CarS of *V. cholerae*. This observation prompted us to designate VVMO6_RS15990 as *carS* and to suggest that the products of *carRS* constitute the CarRS TCS of *V. vulnificus*. To examine the role of CarR of *V. vulnificus* in the PMB resistance, a $\Delta carR$ strain was constructed and its growth was compared with that of the wild-type strain in the absence or presence of PMB. In the absence of PMB, the growth of the $\Delta carR$ strain was similar to that of the wild-type strain (Fig. 1A). However, the growth of the $\Delta carR$ strain was significantly retarded in the presence of PMB and restored to the wild-type level by complementation (Fig. 1B). These results suggested that the response regulator CarR contributes to the PMB resistance of *V. vulnificus*.

**Transcriptome analysis identified the CarR-regulated genes.** For comprehensive identification of the CarR regulon, the transcriptome changes induced by the *carR* deletion were analyzed by RNA-seq. Compared with the wild-type strain, the $\Delta carR$ strain differentially expressed a total of 28 genes, of which 26 genes were downregulated and 2 genes were upregulated (Table S1 in the supplemental material). The number of the downregulated genes and their overall fold changes were greater than those of the upregulated genes (Table S1), implying that CarR primarily acts as an activator rather than a repressor.

Among the downregulated genes, VVMO6_RS15985 showed the highest fold change (about 400-fold), and the sequentially located VVMO6_RS15980 and VVMO6_RS15975 also showed similar fold changes by the *carR* deletion (Fig. 2). The proteins encoded by VVMO6_RS15985, VVMO6_RS15980, and VVMO6_RS15975 are involved in the LPS modification system, showing homology with diacylglycerol kinase DgkA, phosphoethanolamine transferase EptA, and phosphoglycerol transferase I MdoB, respectively. These results suggested that the three genes VVMO6_RS15985, VVMO6_RS15980, and VVMO6_RS15975 (*dgkA*, *eptA*, and *mdoB*, respectively) are activated by CarR in an operon, hereinafter named the *eptA* operon. Besides the *eptA* operon, VVMO6_RS21740, encoding the TolC-like outer membrane channel protein TolCV2, was also differentially expressed by the *carR* deletion (Fig. 2). VVMO6_RS21740 (*tolCV2*) is the second ORF of an operon containing four additional ORFs, VVMO6_RS21745, VVMO6_RS21735, VVMO6_RS21730, and VVMO6_RS21725, hereinafter named the *tolCV2* operon (Table S1). The *tolCV2* operon is predicted to encode a tripartite efflux pump which consists of the inner membrane transporter, periplasmic

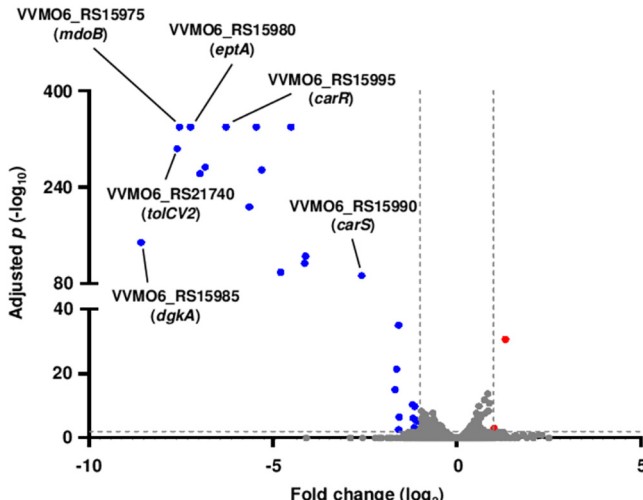

**FIG 2** Analysis of transcriptomes differentially expressed by the *carR* deletion. The genes differentially expressed by the *carR* deletion are visualized in a volcano plot. The gray dashed lines represent the cutoffs for differential expression of fold change ≥2 with adjusted *P* value <0.01. The blue and red dots indicate the genes differentially downregulated and upregulated by the *carR* deletion, respectively.

membrane fusion protein, and outer membrane channel protein (21). In addition, the *carRS* operon is also downregulated by the *carR* deletion, suggesting that CarR activates its own expression (Fig. 2). Altogether, the transcriptome analysis revealed that CarR strongly activates the expression of the *eptA*, *tolCV2*, and *carRS* operons.

**CarR develops the PMB resistance through the activation of the *eptA* operon.** To validate the RNA-seq results, the transcript levels of the CarR regulon in the Δ*carR* strain were compared with those in the wild-type strain. Consistent with the RNA-seq results, the transcript levels of *eptA*, *tolCV2*, and *carR* were extremely lower in the Δ*carR* strain than those in the wild-type strain (Fig. 3A). The results indicated again that CarR is a strong activator for the *eptA*, *tolCV2*, and *carRS* operons.

It has been reported that both the LPS modification system and the tripartite efflux pump are involved in the PMB resistance of other bacteria (22–24). Therefore, further investigations were performed to figure out whether the *eptA* and/or *tolCV2* operons directly confer the PMB resistance on *V. vulnificus*. The Δ*eptA* strain was first generated, and its MIC of PMB was compared with that in the wild-type strain. The Δ*eptA* strain

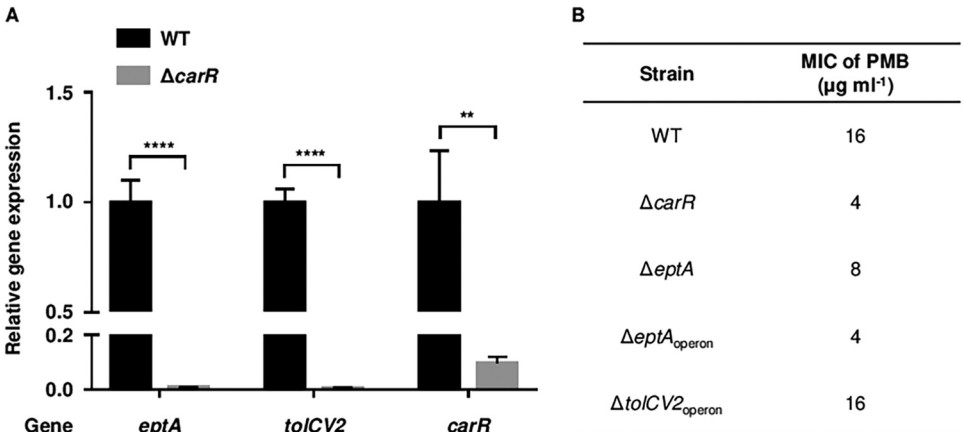

**FIG 3** The genes regulated by CarR and their effects on PMB resistance. (A) Total RNAs were isolated from the *V. vulnificus* strains grown to an $A_{600}$ of 0.5. The transcript levels of *eptA*, *tolCV2*, and *carR* were determined by qRT-PCR, and the transcript level of each gene in the wild type was set at 1. Error bars represent the SD from three independent experiments. Statistical significance was determined by Student's *t* test. **, $P < 0.005$; ****, $P < 0.0001$. (B) The MICs of PMB in the *V. vulnificus* strains were determined using the broth microdilution method. WT, wild type; Δ*carR*, *carR* mutant; Δ*eptA*, *eptA* mutant; Δ*eptA*$_{operon}$, a mutant deficient in the *eptA* operon; Δ*tolCV2*$_{operon}$, a mutant deficient in the *tolCV2* operon.

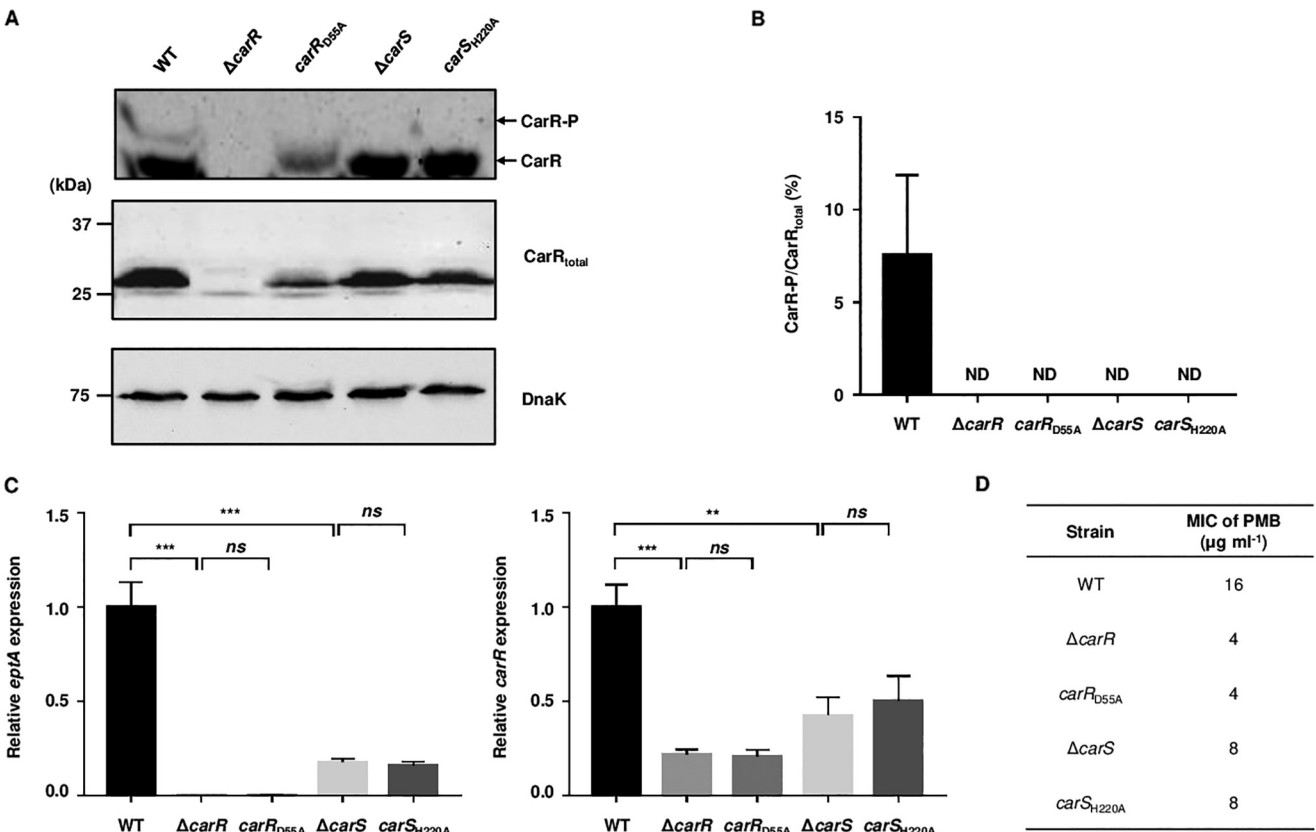

**FIG 4** Roles of D55 of CarR and H220 of CarS in CarR phosphorylation, gene regulation, and PMB resistance. (A to C) Total proteins and RNAs were isolated from the *V. vulnificus* strains grown to an $A_{600}$ of 0.5. (A) The phosphorylation status of CarR was determined by Western blot analysis using Phos-tag SDS-PAGE gel (Wako). The cellular levels of $CarR_{total}$ and DnaK (as an internal control) were determined by Western blot analysis using standard SDS-PAGE gel. Molecular size markers (Bio-Rad) are shown in kDa. (B) The intensities of the protein bands were quantified using Image Lab software (Bio-Rad). ND, not detected. (C) The *eptA* and *carR* transcript levels were determined by qRT-PCR, and the transcript level of each gene in the wild type was set at 1. (D) The MICs of PMB in the *V. vulnificus* strains were determined using the broth microdilution method. Error bars represent the SD from three independent experiments. Statistical significance was determined by Student's *t* test. **, $P < 0.005$; ***, $P < 0.001$; *ns*, not significant; WT, wild type; Δ*carR*; *carR* mutant; *carR*$_{D55A}$, a mutant producing CarR$_{D55A}$; Δ*carS*, *carS* mutant; *carS*$_{H220A}$, a mutant producing CarS$_{H220A}$.

showed a 2-fold lower MIC than the wild-type strain (Fig. 3B), indicating that EptA increases the PMB resistance of *V. vulnificus*. Then, the *dgkA* and *mdoB* genes were further deleted in the Δ*eptA* strain to generate the Δ*eptA*$_{operon}$ strain deficient in the whole *eptA* operon. The MIC of PMB in the Δ*eptA*$_{operon}$ strain was further decreased to a level identical to that in the Δ*carR* strain (Fig. 3B). On the other hand, deletion of the *tolCV2* operon did not affect the PMB resistance of *V. vulnificus* (Fig. 3B). Altogether, these results suggested that CarR develops the PMB resistance of *V. vulnificus* by activating *eptA* expression and that two additional genes in the *eptA* operon, *dgkA* and *mdoB*, are required for complete development of the antibiotic resistance.

**CarR phosphorylation is essential for the gene regulation and PMB resistance.** CarR and CarS of *V. vulnificus* contain a conserved Asp (D55) and His (H220) residue, respectively, which are expected to be phosphorylated (Fig. S1). To examine the roles of the two residues in the CarR phosphorylation, the *carR*$_{D55A}$ strain, producing a mutant CarR with Ala substitution of D55 (CarR$_{D55A}$), and the *carS*$_{H220A}$ strain, producing a mutant CarS with Ala substitution of H220 (CarS$_{H220A}$), were generated, and the phosphorylation status of CarR in the *V. vulnificus* strains was evaluated. Phosphorylated CarR (CarR-P) was clearly detected in the wild-type strain, and its cellular level accounted for about 8% of that of total CarR (CarR$_{total}$) under our test condition (Fig. 4A and B). However, CarR-P was not detected in the *carR*$_{D55A}$ strain (Fig. 4A and B), indicating that D55 is the phosphorylation site of CarR. CarR-P in the Δ*carS* and *carS*$_{H220A}$ strains was also undetectable by immunoblotting (Fig. 4A and B), indicating that H220 of CarS is obviously involved in the CarR phosphorylation.

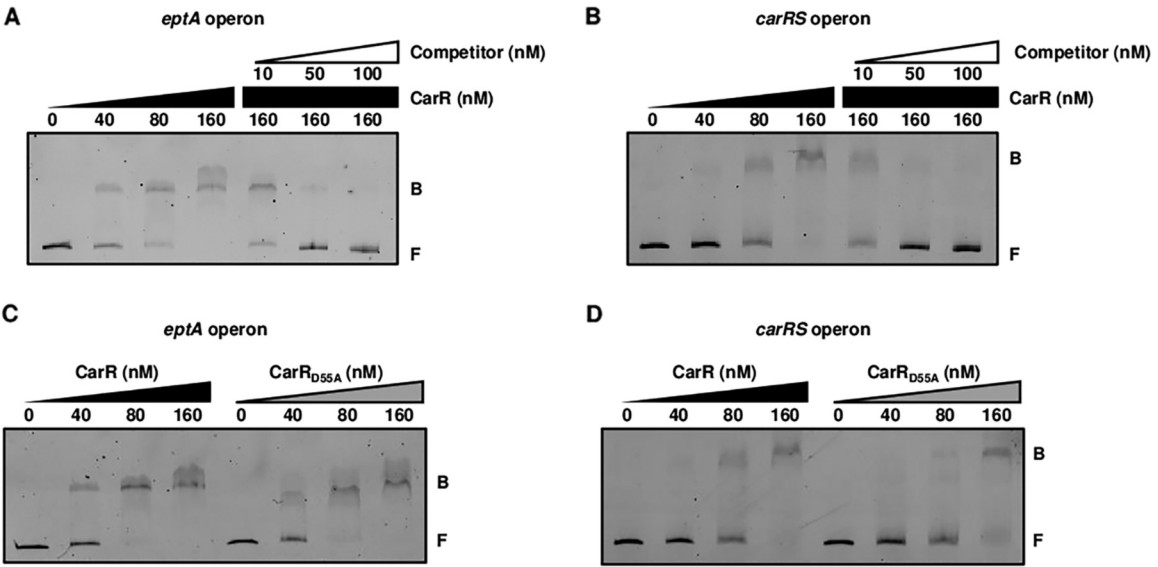

**FIG 5** Direct binding of CarR and CarR$_{D55A}$ to the upstream regions of the CarR regulon. (A to D) The 6-FAM-labeled DNA probes (5 nM) for the upstream regions of the *eptA* operon (A and C) and the *carRS* operon (B and D) were incubated with increasing amounts of CarR or CarR$_{D55A}$ as indicated. (A and B) For competition analysis, the same but unlabeled DNA probes were used as self-competitors, and various amounts of self-competitors were added to the reaction mixtures before the addition of CarR as indicated. B, bound DNA; F, free DNA.

Then, the effects of the CarR phosphorylation on the gene regulation and PMB resistance were examined. The expression levels of *eptA* in the *carR*$_{D55A}$ and *carS*$_{H220A}$ strains were comparable with those in the Δ*carR* and Δ*carS* strains, respectively (Fig. 4C). Similarly, the expression levels of *carR* in the *carR*$_{D55A}$ and *carS*$_{H220A}$ strains were also close to those in the Δ*carR* and Δ*carS* strains, respectively (Fig. 4C). Additionally, the MICs of PMB in the *carR*$_{D55A}$ and *carS*$_{H220A}$ strains were the same as those in the Δ*carR* and Δ*carS* strains, respectively (Fig. 4D). The combined results indicated that CarR phosphorylation is essential for the activation of the *eptA* and *carR* expression, and thus, the development of PMB resistance. Altogether, CarS phosphorylates CarR, and the resulting CarR-P, an active form of CarR, activates the expression of its downstream genes, leading to the PMB resistance of *V. vulnificus*.

**CarR directly binds to the upstream regions of the *eptA* and *carRS* operons.** To identify whether CarR directly regulates the expression of the *eptA* and *carRS* operons, CarR binding to their upstream regions was examined by electrophoretic mobility shift assays (EMSAs). The addition of CarR to each labeled DNA probe, encompassing the upstream region of either the *eptA* or *carRS* operon, resulted in a single retarded band of a CarR-DNA complex in a CarR concentration-dependent manner. Additionally, the same but unlabeled DNA probes competed for binding to CarR in a dose-dependent manner (Fig. 5A and B). These results indicated that CarR directly binds to the upstream regions of the *eptA* and *carRS* operons. Then, the binding of CarR$_{D55A}$ to the same upstream regions was compared with that of CarR. Interestingly, the binding affinity of CarR$_{D55A}$ to each upstream region was similar to that of CarR (Fig. 5C and D), suggesting that CarR does not require phosphorylation to bind to the upstream regions of the *eptA* and *carRS* operons.

To identify the specific binding sequences for CarR in the upstream regions of the *eptA* and *carRS* operons, DNase I protection assays were performed using the same DNA probes. The addition of CarR resulted in the protection of a single binding site of 35-bp and 27-bp in the upstream regions of the *eptA* and *carRS* operons, respectively (Fig. 6A and B). Alignment of the DNA sequences protected by CarR revealed a consensus motif, GACAN$_6$TACA (Fig. 6C), which is proposed as a putative CarR binding sequence. Combined with the EMSA results, these results suggested that, whether phosphorylated or not, CarR directly binds to the specific sequences in the upstream regions of the *eptA* and *carRS* operons.

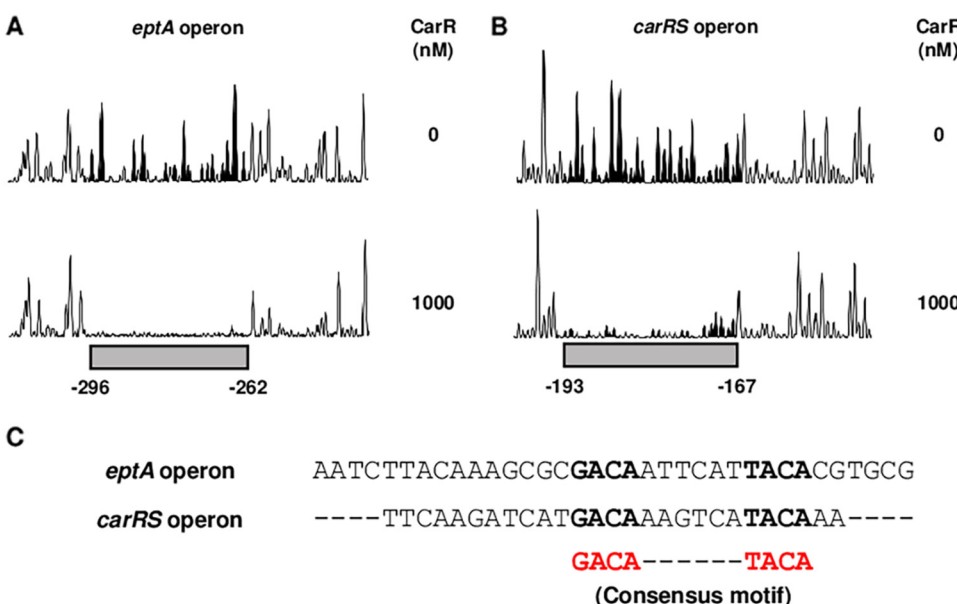

**FIG 6** Specific and conserved binding sequences of CarR. (A and B) The 6-FAM-labeled DNA probes (40 nM) for the upstream regions of the *eptA* operon (A) and the *carRS* operon (B) were incubated with or without CarR (1000 nM) and then digested with DNase I. Each region protected by CarR is represented by a gray box. Nucleotides are numbered relative to the first base of the ORF for *dgkA* of the *eptA* operon (A) and *carR* of the *carRS* operon (B). (C) The sequences of the CarR-protected regions are aligned, and the consensus motif for the CarR binding is shown in red. The bases that match the consensus motif are shown in boldface.

**The CarRS TCS responds to multiple signals related to host environments.** It has been widely known that TCSs respond to certain environmental signals and alter their own activation state, resulting in changes in their expression levels. To identify the specific signals to which the CarRS TCS responds, the levels of the *carRS* promoter ($P_{carRS}$) activities were examined in the *V. vulnificus* strains under different host environmental stresses. For this purpose, the $P_{carRS}$ activities were determined by measuring the cellular luminescence of the *V. vulnificus* strains carrying the $P_{carRS}$-*luxCDABE* reporter plasmid. The $P_{carRS}$ activity of the wild-type strain was significantly increased by the addition of PMB and decreased by the addition of divalent cations, including $Mg^{2+}$ and $Ca^{2+}$, and bile salts to the culture medium, whereas the $P_{carRS}$ activities of the $\Delta carR$ and $\Delta carS$ strains were not altered by exposure to these environmental stresses (Fig. 7A to C). Meanwhile, the $P_{carRS}$ activities of the $\Delta carR$ and $\Delta carS$ strains, as well as the wild-type strain, were elevated by the acidic pH in the culture medium. However, the $P_{carRS}$ activity of the wild-type strain showed a greater increment than those of the $\Delta carR$ and $\Delta carS$ strains (Fig. 7D). Collectively, these results suggested that the CarRS TCS alters its activation state by responding to host-related signals such as PMB, divalent cations, bile salts, and pH change.

**CarR is associated with resistance to bile salts and acidic pH.** To investigate the physiological role of the CarRS TCS responding to the host-related signals, the growth of the wild-type and $\Delta carR$ strains was monitored under various environmental stresses. The growth of the wild type was not affected by the *carR* deletion in the extremely low concentration of $Mg^{2+}$ (Fig. S2A). However, the growth rate of the wild type was altered by the *carR* deletion in the presence of bile salts or in the acidic pH (Fig. S2B and C). Therefore, the surviving *V. vulnificus* cells exposed to bile salts or acidic pH were further quantitated. Compared with the wild type, the survival of the $\Delta carR$ strain was significantly enhanced in the presence of bile salts but reduced at acidic pH (Fig. 8). Altogether, these results suggested that CarR affects the ability of *V. vulnificus* to survive and overcome host stresses including bile salts and acidic pH.

## DISCUSSION

In this study, the CarRS TCS that develops the PMB resistance of *V. vulnificus* was newly identified (Fig. 1). The response regulator CarR activates the expression of the *eptA*

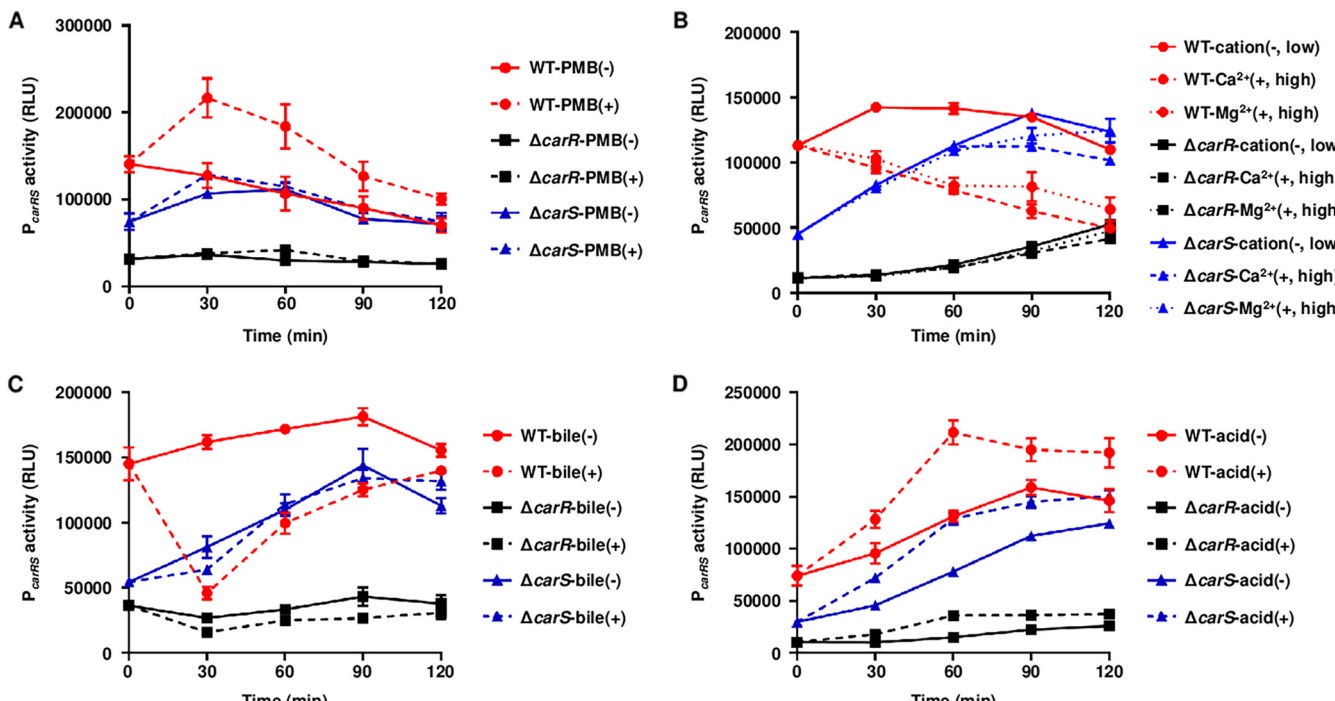

**FIG 7** Effects of various host-related stresses on the P*carRS* activity. (A to D) The *V. vulnificus* strains carrying reporter plasmid pDH2207 with P*carRS* fused to promoterless *luxCDABE* were grown to an $A_{600}$ of 0.5 and further incubated for 120 min with (+) or without (−) stresses as follows: PMB, 1 $\mu$g mL$^{-1}$ (A); CaCl$_2$ or MgCl$_2$, 10 mM (B); bile salts, 0.03% (wt/vol) (C); acidic pH (pH 5.7) (D). The cellular luminescence and growth ($A_{600}$) were measured at time intervals as indicated. RLU was calculated by dividing the luminescence by the $A_{600}$. Error bars represent the SD from three independent experiments. RLU, relative luminescence unit; WT, wild type carrying pDH2207; Δ*carR*, *carR* mutant carrying pDH2207; Δ*carS*, *carS* mutant carrying pDH2207.

operon consisting of three genes, *dgkA*, *eptA*, and *mdoB*, thereby providing *V. vulnificus* with the PMB resistance (Fig. 2 and 3). It has been well known that EptA adds a phosphoethanolamine moiety to lipid A and confers PMB resistance to other *Vibrio* species including *V. cholerae* and *V. parahaemolyticus* (25–27). However, the *eptA* expression is not regulated by the CarRS TCS in *V. cholerae* (26). In *V. parahaemolyticus*, EptA increases survival within macrophage cells (27), suggesting that LPS modification induces resistance to host immune clearance. Meanwhile, the expression of the *tolCV2* operon is also positively regulated by CarR, showing an expression fold change similar to that of the

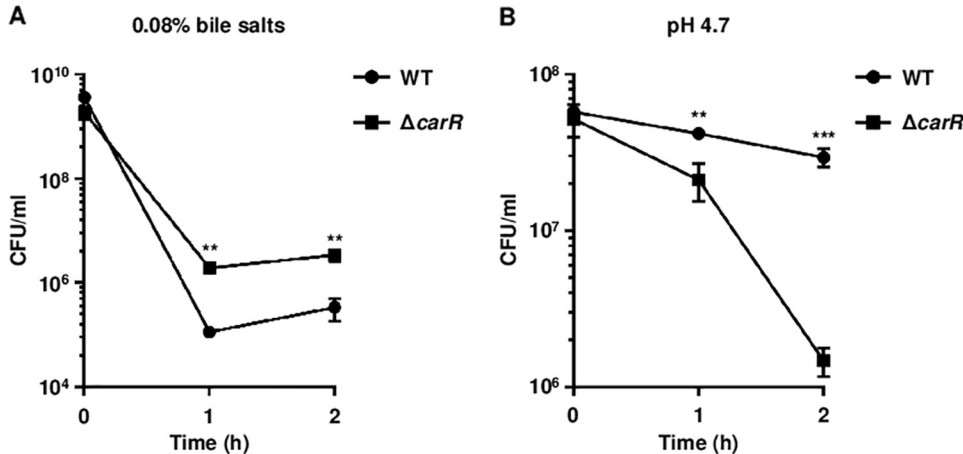

**FIG 8** Survival of *V. vulnificus* exposed to bile salts and acidic pH. (A and B) The *V. vulnificus* strains grown to an $A_{600}$ of 0.5 were further incubated for 2 h in LBS containing 0.08% (wt/vol) bile salts (A) or LBS adjusted to pH 4.7 (B). The surviving cells were plated and then enumerated at each time point. Error bars represent the SD from three independent experiments. Statistical significance was determined by Student's *t* test. **, $P < 0.005$; ***, $P < 0.001$; WT, wild type; Δ*carR*, *carR* mutant.

*eptA* operon (Table S1). However, the *tolCV2* operon does not affect the PMB resistance of *V. vulnificus* (Fig. 3B). Although the biological role of *tolCV2* has not yet been understood, a previous study using *in vivo* expression technology (IVET) revealed that the *tolCV2* expression is specifically induced inside the host tissue, leading to the proposal of TolCV2 as a putative virulence factor of *V. vulnificus* (28). Thus, the regulation of *tolCV2* by the CarRS TCS could be crucial for the survival and virulence of *V. vulnificus* in the host environment, but the details should be investigated in future studies. Altogether, the CarRS TCS, which highly activates the expression of the *eptA* and *tolCV2* operons, could enhance not only the PMB resistance but also the pathogenicity of *V. vulnificus* during infection.

In general, a sensor kinase phosphorylates its cognate response regulator, and the phosphorylation of the response regulator induces structural change, facilitates dimerization, and in turn, increases DNA-binding affinity (29). Thus, only a phosphorylated response regulator is considered to bind to the regulatory region of the target gene and to modulate its expression. Similarly, CarS phosphorylates CarR, and the resulting CarR-P increases the expression of *eptA* and *carR*, as well as the resistance to PMB (Fig. 4). In addition, EMSAs and DNase I protection assays identified that CarR directly and specifically binds to the upstream regions of the *eptA* and *carRS* operons (Fig. 5 and 6). Interestingly, however, CarR does not require phosphorylation for DNA binding (Fig. 5C and D), indicating another role of CarR phosphorylation besides enhancing DNA-binding affinity. One possible explanation for the CarR-P-dependent gene regulation is that only CarR-P, and not CarR, interacts with RNA polymerase after DNA binding and then promotes transcription of the downstream genes. Indeed, a few response regulators, such as OmpR and BvgA, have been reported to bind DNA regardless of phosphorylation but to activate the target gene expression only when phosphorylated (30–32).

The P$_{carRS}$ activity of *V. vulnificus* increases upon exposure to PMB and acidic pH, while it decreases in the presence of high levels of divalent cations and bile salts (Fig. 7). This indicates that the presence of PMB, low levels of the divalent cations, and acidic pH are environmental signals for the activation of the CarRS TCS, but the presence of bile salts is a signal for the inactivation of the CarRS TCS. Furthermore, CarR has a positive effect on not only the PMB resistance but also the survival at acidic pH, while exhibiting a negative effect on the survival in the presence of bile salts (Fig. 8). Thus, it is tempting to speculate that the CarRS TCS provides *V. vulnificus* with the benefit of surviving within the host by appropriately being activated or inactivated in response to each environmental signal. Our previous RNA-seq analyses showed that the expression of the *carRS* operon is downregulated upon exposure to nitric oxide (33) or murine blood (34). These results indicated that nitrosative stress or blood component(s) could be another signal for the inactivation of the CarRS TCS. Collectively, the CarRS TCS modulates its own state between activation and inactivation upon sensing multiple host-related signals in different niches, contributing to the pathogenesis of *V. vulnificus*.

In summary, the CarRS TCS of *V. vulnificus*, which is responsible for the PMB resistance, has been characterized. CarR activates the expression of the *eptA* and *carRS* operons by directly and specifically binding to their upstream regions, and CarR phosphorylation by CarS is important for the gene regulation. Notably, the CarRS TCS alters its activation state by responding to host-related stresses including PMB, divalent cations, bile salts, and pH change. Furthermore, CarR determines the resistance of *V. vulnificus* to bile salts and acidic pH among the stresses as well as PMB. Altogether, the CarRS TCS could contribute to the optimal fitness of *V. vulnificus* during infection by recognizing and responding to multiple host environmental signals.

## MATERIALS AND METHODS

**Strains, plasmids, and culture conditions.** The strains and plasmids used in this study are listed in Table S2. Unless otherwise noted, the *V. vulnificus* strains were grown in Luria-Bertani medium supplemented with 2% (wt/vol) NaCl (LBS) at 30°C. When required, antibiotics were added to the medium at the following concentrations: ampicillin, 100 $\mu$g mL$^{-1}$; kanamycin, 100 $\mu$g mL$^{-1}$; and chloramphenicol, 3 $\mu$g mL$^{-1}$. The growth of the *V. vulnificus* strains was monitored spectrophotometrically at 600 nm

($A_{600}$). When indicated, overnight cultures of the *V. vulnificus* strains were diluted to an $A_{600}$ of 0.01 and then used to inoculate LBS with or without PMB (10 $\mu$g mL$^{-1}$).

**Identification of *V. vulnificus carR*.** To identify genes responsible for the PMB resistance, a previously constructed random transposon mutant library of *V. vulnificus* was used (35). From the transposon mutants, a mutant exhibiting a decreased growth rate in LBS containing PMB (50 $\mu$g mL$^{-1}$) was screened. A DNA fragment flanking the transposon insertion was amplified by PCR as described previously (36), and a search for homology between the sequence of the resulting fragment and the *V. vulnificus* MO6-24/O genome (GenBank assembly accession number GCF_000186585.1) singled out *carR*.

**Generation and complementation of mutants.** For construction of isogenic deletion mutants, target genes were inactivated *in vitro* by deletion of each ORF using the PCR-mediated linker-scanning method as described previously (37). Briefly, appropriate pairs of primers were used for amplification of the deleted ORF fragment (Table S3), and the resulting fragment was ligated into SpeI-SphI-digested pDM4 (38). *Escherichia coli* S17-1 $\lambda pir$ containing pDM4 with the deleted ORF fragment was used as a conjugal donor to *V. vulnificus* MO6-24/O to generate each deletion mutant (Table S2). The conjugation and isolation of the transconjugants were conducted as described previously (39).

To complement the *carR* deletion, the *carR* ORF was amplified by CARRC-F and CARRC-R (Table S3). The resulting fragment was cloned into pJK1113 (40) under the arabinose-inducible promoter $P_{BAD}$ to generate pTY21091 (Table S2). pJK1113 and pTY21091 were transferred into the appropriate strains by conjugation as described above.

**Transcriptome analysis and quantitative reverse transcription-PCR (qRT-PCR).** The wild-type and $\Delta carR$ strains with two biological replicates were grown to an $A_{600}$ of 0.5, and the total RNAs were isolated using an RNeasy minikit (Qiagen, Valencia, CA). Strand-specific cDNA libraries were constructed and sequenced using HiSeq 2500 (Illumina, San Diego, CA) by CJ Bioscience (Seoul, South Korea) as described previously (41). The raw sequencing reads were mapped to the *V. vulnificus* MO6-24/O reference genome (GenBank assembly accession number GCF_000186585.1). The expression level of each gene was calculated as reads per kilobase of transcript per million mapped sequence reads (RPKM) using EDGE-pro (Estimated Degree of Gene Expression in PROkaryotes) version 1.3.1 (42). The RPKM values were normalized and analyzed statistically as described previously (43) to identify the differentially expressed genes (fold change $\geq$2 with adjusted *P* value <0.01).

For qRT-PCR, cDNA was synthesized from 500 ng of the total RNAs using PrimeScript RT master mix (TaKaRa, Tokyo, Japan). Real-time PCR amplification of the cDNA was performed using a CFX96 real-time PCR detection system (Bio-Rad, Hercules, CA) with pairs of specific primers (Table S3) as described previously (44). The relative expression levels of the transcripts were calculated using the expression level of *rrsH* as an internal reference for normalization.

**Antimicrobial susceptibility test.** The MICs of PMB in the *V. vulnificus* strains were determined by the broth microdilution method according to the CLSI (Clinical and Laboratory Standards Institute) guidelines (45). The MIC of PMB in *E. coli* ATCC 25922 was used as the quality control.

**Construction of mutants producing CarR$_{D55A}$ and CarS$_{H220A}$.** The D55 of CarR and the H220 of CarS were substituted with Ala to examine their role in CarR phosphorylation. The mutated *carR*$_{D55A}$ and *carS*$_{H220A}$ regions were constructed using the PCR-mediated linker-scanning method as described above. To construct the mutated *carR*$_{D55A}$ region, the mutagenic primers CARRPM-R1 and CARRPM-F2 were designed to carry the substitution of D55 of CarR (G<u>AC</u>) with Ala (G<u>CT</u>). Then, primer pairs CARRD-F1 and CARRPM-R1 and CARRPM-F2 and CARRD-R2 were used for amplification of the 5′ amplicon and 3′ amplicon of *carR*$_{D55A}$, respectively (Table S3). Similarly, the mutagenic primers CARSPM-R1 and CARSPM-F2 were designed to carry the substitution of H220 of CarS (C<u>AT</u>) with Ala (G<u>CT</u>) to construct the mutated *carS*$_{H220A}$ region (Table S3). Then, primer pairs CARSD-F1 and CARSPM-R1 and CARSPM-F2 and CARSD-R2 were used for amplification of the 5′ amplicon and 3′ amplicon of *carS*$_{H220A}$, respectively. The resulting 5′ amplicon and 3′ amplicon of either *carR*$_{D55A}$ or *carS*$_{H220A}$ were ligated into SpeI-SphI-digested pDM4 to generate pGR2101 or pGR2102. *E. coli* S17-1 $\lambda pir$ containing either pGR2101 or pGR2102 was used as a conjugal donor to the $\Delta carR$ or $\Delta carS$ strain to generate the *carR*$_{D55A}$ or *carS*$_{H220A}$ strain (Table S2). The conjugation and isolation of the transconjugants were conducted as described above.

**Protein purification and Western blot analysis.** To overexpress CarR and CarR$_{D55A}$, the *carR* and *carR*$_{D55A}$ genes were amplified by PCR using CARRP-F and CARRP-R from the wild-type and *carR*$_{D55A}$ strains, respectively (Table S3). Each resulting fragment was cloned into pET-28a(+) (Novagen, Madison, WI) to generate pTY21092 and pDH2214 (Table S2). The His$_6$-tagged CarR and CarR$_{D55A}$ were expressed in *E. coli* BL21(DE3) and purified by affinity chromatography (Qiagen). The purified His$_6$-tagged CarR was used to raise mouse anti-CarR polyclonal antibody (AbClon, Seoul, South Korea).

For Western blot analysis, the *V. vulnificus* cells were harvested by centrifugation and lysed using B-PER bacterial protein extraction reagent with enzymes (Thermo Fisher Scientific, Waltham, MA). After removal of the cell debris by centrifugation, the protein levels of CarR and DnaK were detected using mouse anti-*V. vulnificus* CarR antibody and mouse anti-*E. coli* DnaK antibody (Enzo Life Science, Farmingdale, NY, USA) as described previously (46). The phosphorylated CarR was separated from the unphosphorylated CarR using 10% SuperSep Phos-tag SDS-PAGE gel (Wako, Osaka, Japan) (47). The cellular levels of CarR-P and CarR$_{total}$ were quantified using Image Lab software (Bio-Rad).

**EMSAs and DNase I protection assays.** For EMSA, the 500-bp DNA fragment of the upstream region of the *eptA* operon was amplified by PCR using unlabeled PEPTA-F and 6-carboxyfluorescein (6-FAM)-labeled PEPTA-R as the primers (Table S3). Similarly, the 375-bp DNA fragment of the upstream region of the *carRS* operon was amplified by PCR using unlabeled PCARR-F and 6-FAM-labeled PCARR-R as the primers (Table S3). Each 6-FAM-labeled DNA probe was incubated with different amounts of purified His$_6$-tagged CarR or CarR$_{D55A}$ for 30 min at 25°C in a 20-$\mu$L reaction mixture containing 1$\times$ CarR binding

buffer (40 mM Tris-Cl [pH 8.0], 50 mM NaCl, 1 mM dithiothreitol [DTT], 100 $\mu$M EDTA, 0.1 $\mu$g $\mu$L$^{-1}$ bovine serum albumin [BSA]) and 0.05 $\mu$g of poly(dI-dC) (Sigma-Aldrich, St. Louis, MO). For competition analysis, the same but unlabeled DNA probe was used as a self-competitor. Electrophoretic analyses of the protein-DNA complexes were performed as described previously (48).

For the DNase I protection assay, the same 6-FAM-labeled DNA probes for the upstream regions of the *eptA* and *carRS* operons were used. Each 6-FAM-labeled DNA probe was incubated with His$_6$-tagged CarR for 30 min at 25°C in a 20-$\mu$L reaction mixture containing 1× CarR binding buffer and 0.05 $\mu$g of poly(dI-dC) (Sigma-Aldrich). The CarR-DNA complexes were digested with DNase I as described previously (34). The digested DNA products were precipitated with ethanol, eluted in distilled water, and analyzed using an ABI 3730xl DNA analyzer (Applied Biosystems, Foster City, CA) with Peak Scanner software version 1.0 (Applied Biosystems).

**Construction of the P$_{carRS}$-luxCDABE transcriptional fusion.** The 371-bp *carR* upstream region (positions from −314 to +57 relative to the first base of the ORF for *carR*) was amplified using primer pair PcarRX-F and PcarRX-R (Table S3) and fused to SacI-SpeI-digested pBBR-lux carrying the promoterless *luxCDABE* genes to generate pDH2207 (Table S2) (49). pDH2207 was transferred into the *V. vulnificus* strains by conjugation as described above. Each strain was grown to an $A_{600}$ of 0.5, and its cellular luminescence and growth ($A_{600}$) were monitored using a Tecan Spark microplate reader (Tecan, Mannedorf, Switzerland) upon exposure to various host-related stresses: PMB (1 $\mu$g mL$^{-1}$), CaCl$_2$ (10 mM), MgCl$_2$ (10 mM), bile salts (0.03%), and acidic pH (pH 5.7). Relative luminescence units (RLUs) were calculated by dividing the luminescence by the $A_{600}$ (41).

**Viability assay.** The viability of the *V. vulnificus* strains under host environmental stresses was determined by the spot plating method. To measure the growth under different Mg$^{2+}$ levels, the *V. vulnificus* strains grown overnight in LBS were diluted 1:100 in M9 minimal medium containing 2 mM or 50 $\mu$M MgSO$_4$ and further incubated for 9 h. To measure the effect of bile salts or acidic pH on growth, the *V. vulnificus* strains grown to an $A_{600}$ of 0.5 were further incubated in LBS containing 0.08% (wt/vol) bile salts or LBS adjusted to pH 4.7 for 2 h. Equal volumes of the *V. vulnificus* strains were serially 10-fold diluted, plated onto LBS agar, and incubated for 14 h. When indicated, the numbers of the surviving *V. vulnificus* cells were enumerated at each time point.

**Data analysis.** Average values and standard deviations (SD) were calculated from at least three independent experiments. Statistical analysis was performed with Student's *t* test using GraphPad Prism 7.0 (GraphPad Software, San Diego, CA).

**Data availability.** All raw transcriptome data have been deposited in the NCBI BioProject database under accession number PRJNA905907.

## SUPPLEMENTAL MATERIAL

Supplemental material is available online only.
**SUPPLEMENTAL FILE 1**, DOCX file, 0.8 MB.

## ACKNOWLEDGMENTS

This work was supported by the National Research Foundation of Korea (NRF) grant funded by the Korea government (MSIT) (grant No. 2023R1A2C1002968 to S.H.C.), the Cooperative Research Program for Agriculture Science and Technology Development (project No. PJ016298 to S.H.C.), Rural Development Administration, Republic of Korea, and the Creative-Pioneering Researchers Program through Seoul National University to Y.-J.B.

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
