## [Reviewer comments · Microbiology Spectrum]

Microbiology Spectrum

CarRS two-component system essential for polymyxin B resistance of *Vibrio vulnificus* responds to multiple host environmental signals

Duhyun Ko, Dayoung Sung, Tae Young Kim, Garam Choi, Ye-Ji Bang, and Sang Ho Choi

Corresponding Author(s): Ye-Ji Bang and Sang Ho Choi, Seoul National University

Review Timeline:

Submission Date:	January 20, 2023
Editorial Decision:	March 17, 2023
Revision Received:	May 11, 2023
Accepted:	May 15, 2023

Editor: Sandeep Tamber

Reviewer(s): The reviewers have opted to remain anonymous.

Transaction Report:

DOI: <https://doi.org/10.1128/spectrum.00305-23>

March 17, 2023

Prof. Sang Ho Choi
Seoul National University
Department of Agricultural Biotechnology
College of Agriculture and Life Science
Shillimdong, Gwanakgu
Seoul 08826
Korea (South), Republic of

Re: Spectrum00305-23 (CarRS two-component system essential for polymyxin B resistance of *Vibrio vulnificus* responds to multiple host environmental signals)

Dear Prof. Sang Ho Choi:

Link Not Available

Sincerely,

Sandeep Tamber

Journals Department
Editor comments:

- L30 - please correct. Fulminating refers to the disease symptom, not the pathogen
- L87 - *V. vulnificus* causes more than just gastroenteritis
- In the discussion can you discuss the role *eptA* plays in LPS modification and compare your findings in *vulnificus* with other *Vibrio* species such as cholera
- Are *carRS* homologues widespread in other *Vibrio* spp.?

Reviewer comments:

Reviewer #1 (Comments for the Author):

Authors describes biological roles of the two-component system CarRS in the pathogenicity of the human pathogen *Vibrio vulnificus*. They showed the regulation of target genes by CarRS, which are associated with a survival in host, in molecular genetic levels.

Overall the story and experimental design and results are quite straightforward, and has a scientific significance.

Reviewer #2 (Comments for the Author):

In this study, Ko et al. investigated the two component systems, TCS, of *Vibrio vulnificus* which provide resistance to polymyxin B, PMB. Using mutant strains of PMB response regulator CarR and downstream CarR regulated genes, the authors find that CarRS TCS provides *V. vulnificus* with resistance to PMB. The authors show that CarR is a regulator of three identified operons, *eptA*, *tolCV2* and *carRS*. They find that CarR activates *eptA* expression and this activation is required for resistance of PMB. They find that *tolCV2* did not appear to affect the resistance to PMB. Additionally, they show that CarR requires phosphorylation mediated by CarS which leads to downstream activation of *eptA* operon. Finally, the authors investigate how the *carRS* operon responds to host related signals, specifically, cations, bile salts and pH changes. They find that CarRS TCS alters its activation in response to all three signals and that CarR affects the ability for *V. vulnificus* to respond to host stresses. Overall, the experiments are well performed, and conclusions are supported.

Major Comments:

In Figure 4, the authors investigate whether CarR phosphorylation is required for gene regulation and PMB resistance. In Figure 4A, the authors show that *carS* and a *carRD55A* point mutant have reduced phosphorylation compared to the wild type. The authors also find that *carS* and *carSH220A* have lower CarR-P than wild type. The authors conclude that H220 residue of CarS is involved in CarR phosphorylation. However, Figure 4A does not fully support this conclusion. The authors base their conclusions on one representative western blot, which is not very clear. The authors should repeat the experiments in Figure 4A and perform densitometry analysis to strengthen their conclusions.

Minor Comments:

Line 230-231: The authors show that *tolCV2* expression is regulated by CarR but does not affect PMB resistance of *V. vulnificus*. Although the authors state some reasons for this, the authors could expand on how future studies can be performed to understand why *tolCV2* is regulated by CarR.

Line 250-251: Authors state that PcarRS activity increases in low levels of cations. However, the data shows that activity is reduced for *V. vulnificus* in the presence of cations. The authors should explain this discrepancy.

Figure 7: Legend states luminescence and growth (A600) were measured at time intervals. However, there is no y-axis which shows A600 readings. Are RLUs and A600 proportional to one another?

Staff Comments:

Preparing Revision Guidelines

Please return the manuscript within 60 days; if you cannot complete the modification within this time period, please contact me. If you do not wish to modify the manuscript and prefer to submit it to another journal, please notify me of your decision immediately so that the manuscript may be formally withdrawn from consideration by Microbiology Spectrum.

Authors describes biological roles of the two-component system CarRS in the pathogenicity of the human pathogen *Vibrio vulnificus*. They showed the regulation of target genes by CarRS, which are associated with a survival in host, in molecular genetic levels.

Overall the story and experimental design and results are quite straightforward, and has a scientific significance.

Major points.

CarRS has been well studied in the related *Vibrio* species. What is the novelty of manuscript? Authors need to highlight of scientific significance of their finding.

Minor points

What would be a working mechanisms of *eptA*?

What would be a working mode of CarRS in response to signals? Do you have any working model?

Line 110~113 It would be helpful for readers to show amino acid alignment b/w *V. vulnificus* and *V. cholerae*. as a supplementary data.

Line 134, *eptA* operon? Not CarRS regulon?? Are those three ORF (15985, 15980, 15975) linked without any interruptions? Generic map of these three genes would be good as a supplementary data.

Line 158 Do not understand this logic. Data on *dgkA* and *mdoB* mutants??

Fig 5A. What is the upper band in the lane 4? In lanes 5 and 6 in Fig 5C, why bnds are retarded less? Is it possible that the dephosphorylation decreased the affinity to the binding site?

Line 205. P_{carRS} activity??? This appears to be a wrong expression. It would not be the activity of promoter but rather the activity of the activator CarRS, because the transcription exerted by the promoter can also be affected by other factors.

Line 206. Hwy Mg^{2+} and Ca^{2+} ?

Line 321 and thereafter, Not single quotation mark (") but prime mark (').

Line 330 Western → western (lower case)

In this study, Ko et al. investigated the two component systems, TCS, of *Vibrio vulnificus* which provide resistance to polymyxin B, PMB. Using mutant strains of PMB response regulator CarR and downstream CarR regulated genes, the authors find that CarRS TCS provides *V. vulnificus* with resistance to PMB. The authors show that CarR is a regulator of three identified operons, *eptA*, *tolCV2* and *carRS*. They find that CarR activates *eptA* expression and this activation is required for resistance of PMB. They find that *tolCV2* did not appear to affect the resistance to PMB. Additionally, they show that *carR* requires phosphorylation mediated by *carS* which leads to downstream activation of *eptA* operon. Finally, the authors investigate how the *carRS* operon responds to host related signals, specifically, cations, bile salts and pH changes. They find that CarRS TCS alters its activation in response to all three signals and that CarR affects the ability for *V. vulnificus* to respond to host stresses. Overall, the experiments are well performed, and conclusions are supported.

Major Comments:

In Figure 4, the authors investigate whether CarR phosphorylation is required for gene regulation and PMB resistance. In Figure 4A, the authors show that $\Delta carS$ and a $carR_{D55A}$ point mutant have reduced phosphorylation compared to the wild type. The authors also find that $\Delta carS$ and $carS_{H220A}$ have lower CarR-P than wild type. The authors conclude that H220 residue of CarS is involved in CarR phosphorylation. However, Figure 4A does not fully support this conclusion. The authors base their conclusions on one representative western blot, which is not very clear. The authors should repeat the experiments in Figure 4A and perform densitometry analysis to strengthen their conclusions.

Minor Comments:

Line 230-231: The authors show that *tolCV2* expression is regulated by CarR but does not affect PMB resistance of *V. vulnificus*. Although the authors state some reasons for this, the authors could expand on how future studies can be performed to understand why *tolCV2* is regulated by CarR.

Line 250-251: Authors state that PcarRS activity increases in low levels of cations. However, the data shows that activity is reduced for *V. vulnificus* in the presence of cations. The authors should explain this discrepancy.

Figure 7: Legend states luminescence and growth (A_{600}) were measured at time intervals. However, there is no y-axis which shows A_{600} readings. Are RLUs and A_{600} proportional to one another?

Point-by-point response to the Reviewers' comments

(Manuscript #: Spectrum00305-23)

On the comments from Editor

- L30 - please correct. Fulminating refers to the disease symptom, not the pathogen

Response: Thank you for the comment. We have changed “fulminating” to “opportunistic” in the revised manuscript (line 30).

- L87 - *V. vulnificus* causes more than just gastroenteritis

Response: Thank you for the comment. *V. vulnificus* is a human pathogen causing food-borne diseases from mild gastroenteritis to life-threatening septicemia, especially in individuals with underlying predisposed conditions (1, 2). We have now updated the Introduction section including the above description (lines 86-87).

- In the discussion can you discuss the role *eptA* plays in LPS modification and compare your findings in *vulnificus* with other *Vibrio* species such as cholera

Response: Thank you for the suggestion. In this study, we have identified that CarRS contributes to the PMB resistance of *V. vulnificus* by activating the *eptA* expression. EptA decreases the negative charge of bacterial outer membrane by adding phosphoethanolamine to lipid A. This LPS modification prevents the binding of PMB to the membrane, leading to the PMB resistance of many Gram-negative bacteria (3). Although the *eptA* genes are conserved in *Vibrio* species, their regulatory characteristics seem to be different from each other. Indeed, EptA contributes to the PMB resistance of *V. cholerae*, but its expression is not regulated by CarRS (4). We have updated the Discussion section to include above description (lines 228-230).

- Are *carRS* homologues widespread in other *Vibrio* spp.?

Response: Yes. The *CarRS* homologues are widespread in other *Vibrio* species showing high sequence similarity. The *CarRS* homologues of *Vibrio* species and their sequence similarities are listed in Table R1.

On the comments from Reviewer #1

Authors describes biological roles of the two-component system CarRS in the pathogenicity of the human pathogen *Vibrio vulnificus*. They showed the regulation of target genes by CarRS, which are associated with a survival in host, in molecular genetic levels. Overall the story and experimental design and results are quite straightforward, and has a scientific significance.

Response: We appreciate the Reviewer's careful review.

On the comments from Reviewer #2

In this study, Ko et al. investigated the two component systems, TCS, of *Vibrio vulnificus* which provide resistance to polymyxin B, PMB. Using mutant strains of PMB response regulator CarR and downstream CarR regulated genes, the authors find that CarRS TCS provides *V. vulnificus* with resistance to PMB. The authors show that CarR is a regulator of three identified operons, *eptA*, *tolCV2* and *carRS*. They find that CarR activates *eptA* expression and this activation is required for resistance of PMB. They find that *tolCV2* did not appear to affect the resistance to PMB. Additionally, they show that *carR* requires phosphorylation mediated by *carS* which leads to downstream activation of *eptA* operon. Finally, the authors investigate how the *carRS* operon responds to host related signals, specifically, cations, bile salts and pH changes. They find that CarRS TCS alters its activation in response to all three signals and that CarR affects the ability for *V. vulnificus* to respond to host stresses. Overall, the experiments are well performed, and conclusions are supported.

Major Comments:

In Figure 4, the authors investigate whether CarR phosphorylation is required for gene regulation and PMB resistance. In Figure 4A, the authors show that $\Delta carS$ and a *carRD55A* point mutant have reduced phosphorylation compared to the wild type. The authors also find that $\Delta carS$ and *carSH220A* have lower CarR-P than wild type. The authors conclude that H220 residue of CarS is involved in CarR phosphorylation. However, Figure 4A does not fully support this conclusion. The authors base their conclusions on one representative western blot, which is not very clear. The authors should repeat the experiments in Figure 4A and perform densitometry analysis to strengthen their conclusions.

Response: We appreciate the Reviewer's valuable comments. As the Reviewer suggested, we repeated the same experiments to clearly detect the phosphorylated form of CarR (CarR-P) using a Phos-tag SDS-PAGE gel and thus to demonstrate that D55 of CarR and H220 of CarS are essential for the CarR phosphorylation. Due to the difficulties in differentiating CarR and CarR-P using the Phos-tag SDS-PAGE gel, we conducted multiple Western blot analyses by changing several experimental conditions. For example, we tried to optimize the temperature for protein preparation, current and voltage for SDS-PAGE, and even machinery for protein transfer to membrane.

As shown in Figure 4A in the revised manuscript, CarR-P was clearly detected in the wild-type strain but not in the *carRD55A* strain, indicating that D55 of CarR is the phosphorylation site. In addition, the cellular level of CarR-P was decreased in the $\Delta carS$ and *carSH220A* strains to the level not detectable by immunoblotting (Figure 4A in the revised manuscript), indicating that H220 of CarS is obviously involved in the CarR phosphorylation. These results were consistent with our previous results of Figure 4A in the original manuscript. To additionally comply with the Reviewer's comment, we performed densitometry analysis using three independent Western blots and calculated the ratio of the cellular level of CarR-P to that of total CarR. The result showed that the cellular level of CarR-P accounts for about 8% of that of total CarR in the wild-type strain under the condition we tested (Figure 4B in the revised manuscript).

The new results have been incorporated in the Results and Materials and Methods sections of the revised manuscript with appropriate descriptions (lines 166-170, 344-349).

Minor Comments:

Line 230-231: The authors show that *tolCV2* expression is regulated by CarR but does not affect PMB resistance of *V. vulnificus*. Although the authors state some reasons for this, the authors could expand on how future studies can be performed to understand why *tolCV2* is regulated by CarR.

Response: We appreciate the Reviewer for raising this point. As we described in the Discussion section, *TolCV2* is expected to act as a virulence factor of *V. vulnificus*. Because CarRS senses multiple host-related signals such as bile salts and acidic pH (Figure 7), the TCS could contribute to precise expression of *tolCV2* under the host environment. Thus, we are planning to perform several *in vitro* and *in vivo* experiments to understand the role of *TolCV2* during host infection. Additionally, the effect of the *tolCV2* regulation by CarRS on the pathogenicity of *V. vulnificus* will be determined in the future study. We have updated the Discussion section including this as a future direction (lines 237-239).

Line 250-251: Authors state that PcarRS activity increases in low levels of cations. However, the data shows that activity is reduced for *V. vulnificus* in the presence of cations. The authors should explain this discrepancy.

Response: We are sorry for the confusing description in this part. We compared the P_{carRS} activities of the *V. vulnificus* strains by adding either sterile distilled water (solid line; low levels of divalent cations) or 10 mM $CaCl_2$ or 10 mM $MgCl_2$ (dashed or dotted line; high levels of divalent cations) to the culture medium. As shown in Figure 7B, the P_{carRS} activity of wild type is higher in the low levels of cations, compared with that in the high levels of cations. For clear description, we have updated Figure 7B in the revised manuscript.

Figure 7: Legend states luminescence and growth (A600) were measured at time intervals. However, there is no y-axis which shows A600 readings. Are RLUs and A600 proportional to one another?

Response: RLUs are calculated by dividing the luminescence with the A_{600} at time intervals, as mentioned in lines 363-367 in the original manuscript (lines 371-375 in the revised manuscript). For clear description, we have updated the legend of Figure 7 to additionally explain how the RLUs are calculated.

Table R1. The CarRS homologues from different *Vibrio* species

Vibrio species	CarR		CarS	
	Locus tag	Identity ^a	Locus tag	Identity ^b
V. cholerae	N16961_VC01536	64.53%	N16961_VC01537	48.36%
V. parahaemolyticus	DET53_105115	71.49%	DET53_105116	55.63%
V. fluvialis	VFL11327_03163	100%	VFL11327_03164	99.30%
V. navarrensis	DU976_19040	80.60%	DU976_19045	66.19%
V. cideicii	NPU43_002635	80.26%	NPU43_002636	66.43%
V. alginolyticus	C0632_07885	71.93%	C0632_07890	55.63%

^aIdentity compared with CarR of *V. vulnificus*.

^bIdentity compared with CarS of *V. vulnificus*.

References

1. Oliver JD. 2015. The Biology of *Vibrio vulnificus*. Microbiol Spectr 3.
2. Phillips KE, Satchell KJ. 2017. *Vibrio vulnificus*: From Oyster Colonist to Human Pathogen. PLoS Pathog 13:e1006053.
3. Samantha A, Vrielink A. 2020. Lipid A Phosphoethanolamine Transferase: Regulation, Structure and Immune Response. J Mol Biol 432:5184-5196.
4. Herrera CM, Henderson JC, Crofts AA, Trent MS. 2017. Novel coordination of lipopolysaccharide modifications in *Vibrio cholerae* promotes CAMP resistance. Mol Microbiol 106:582-596.

May 15, 2023

Prof. Sang Ho Choi
Seoul National University
Department of Agricultural Biotechnology
College of Agriculture and Life Science
Shillimdong, Gwanakgu
Seoul 08826
Korea (South), Republic of

Re: Spectrum00305-23R1 (CarRS two-component system essential for polymyxin B resistance of *Vibrio vulnificus* responds to multiple host environmental signals)

Dear Prof. Sang Ho Choi:

Your manuscript has been accepted, and I am forwarding it to the ASM Journals Department for publication. You will be notified when your proofs are ready to be viewed.

Sincerely,

Sandeep Tamber
Editor, Microbiology Spectrum
